# The Impact of a Household Food Garden Intervention on Food Security in Lesotho

**DOI:** 10.3390/ijerph17228625

**Published:** 2020-11-20

**Authors:** Corinna May Walsh, Michelle Shannon Fouché, Mariette Nel, Frederik Booysen

**Affiliations:** 1Department of Nutrition and Dietetics, University of the Free State, Bloemfontein 9301, South Africa; m.devilliers@ymail.com; 2Department of Biostatistics, University of the Free State, Bloemfontein 9301, South Africa; gnbsmn@ufs.ac.za; 3School of Economics and Finance, University of the Witwatersrand, Johannesburg 2000, South Africa; frederik.booysen@wits.ac.za

**Keywords:** living poverty index, adequate household food provisioning, dietary diversity

## Abstract

Food insecurity is a challenge in the developing world, where many are finding healthy food inaccessible due to poverty. A pre-test, post-test design was applied to determine the impact of a vegetable gardening intervention in 25 experimental and 25 control households in Lesotho. Information about sociodemographic conditions and indicators of food security was collected by trained fieldworkers. As evidenced by the Living Poverty Index of 2.5, the sample was characterized by high levels of poverty. Although almost no households were scored very low or low using the Months of Adequate Household Food Provisioning (MAHFP) tool, less than half of households were categorized as food-secure. Household Dietary Diversity (HDD) showed infrequent intake of vegetables and fruits and regular intake of fats and sugar. After intervention, the percentage of households with a low HDD score improved significantly in the intervention group (12%) compared to the control group (40%) (95% CI (2.5%; 50.7%)). Despite this, the percentage of households that consumed vegetables during the previous day was still below 30%. Food gardens have the potential to improve availability of food and frequency of vegetable consumption, but harsh environmental conditions need to be considered.

## 1. Introduction

The small mountainous country of Lesotho is populated by about 2.2 million people, with two-thirds living in rural areas. The villages in the lowlands of the country are the most populated [1,2,3]. About 60–70% of the Lesotho population is severely food-insecure [1]. In rural areas, the majority of individuals are dependent on agriculture for survival, which proves challenging as less than 10% of the country’s total area is suitable for growing crops. Land availability is further influenced by urbanization and environmental factors, such as an erratic climate, soil erosion, and climate change [4]. Thus, the number of landless households is steadily increasing [5].

Lesotho is considered to be one of the least developed and poorest countries in the world. This can be attributed to high unemployment rates, a decrease in production from agricultural activities, low life expectancy, and increased child mortality rates. Half of Lesotho’s population lives below the poverty line and depends on remittances to survive [6].

In poor countries, income, food prices, and environmental factors play a larger role in what is eaten than personal choice [7]. Despite these challenges, household agricultural activities form the basis of livelihoods in the majority of households [8].

The main staple of Lesotho is white maize that is imported from South Africa [9,10,11]. Although the country is largely dependent on imports, maize, sorghum, and wheat are grown by smallholders. Beans, peas, and potatoes are also grown by some smallholders [1]. Women from rural households in Lesotho consume fermented and stiff porridge as staples. Protein is mainly obtained from dried beans, while vegetables like onions, cabbage, tomatoes, turnips, and potatoes are consumed almost daily, but are often prepared with fat or oil and eaten in small quantities [9].

The most recent Lesotho Demographic and Health Survey (LDHS) confirmed that a triple burden of malnutrition, characterized by undernutrition (stunting, defined as height-for-age below −2 standard deviations from the WHO mean; and wasting, defined as weight-for-height below −2 standard deviations from the WHO mean), micronutrient deficiency, and overnutrition (overweight and obesity) was evident in 2013 [2]. This is consistent with findings from other low-income countries where undernutrition persists, while overweight and obesity are on the rise [12]. It is not uncommon for stunted and wasted children to live in a household where adults are overweight and/or obese and for these children to become overweight themselves as they become older.

Agricultural wageworkers in Lesotho are predominantly female. Women own land, work alongside wageworkers, and may also assist other women that own land [4]. This system could have developed as a result of the past absence of men in the country when they travelled to surrounding countries in search of work [6,13]. According to the LDHS (2014), Lesotho’s population is 53% female and 47% male, with women heading 36% of all households. The trend of living separated from families for extended periods of time due to distant work and education opportunities was evident in 2013 [2].

Earlier studies in Lesotho reported that, in addition to farming, sharing makes a significant contribution to the livelihoods of people in Lesotho [14]. Lesotho has adopted various forms of sharing. These include the Chief receiving tribute from community members, who is then required to support community members in times of hardship. Members of the community may furthermore informally share resources with each other during times of hardship or help each other farm. Sharecropping has also been adopted. In this system, landowners and providers of inputs, equipment, or services each have designated roles for a farming season. Crops are then shared at the end of the season [14].

Although a number of vegetable garden interventions have been implemented by the Lesotho government and various nongovernmental organizations, nothing has recently been published about their impact. In 1991, a report by Kumar noted that the positive effects of vegetable garden interventions in Lesotho were often lost once donor support ceased [15]. If correctly implemented, household food gardens can provide a supplemental food source, thus improving food security and dietary diversity (DD) [16,17,18,19]. Having a household food garden potentially provides an affordable source of nutritious food, addressing both food access and availability [16,17,18,19] while also contributing to income generation [17,19,20].

In view of the high levels of poverty, malnutrition, and food insecurity in Lesotho, relevant interventions that can address these challenges are urgently required. With this in mind, this study aimed to investigate the potential benefits of household food gardens in Lesotho.

## 2. Materials and Methods

This study formed part of a larger study undertaken by the Centre for Development Support at The University of The Free State from July 2013 to 2015, titled “Household food gardens: effective and sustainable impact mitigation response to the HIV and AIDS epidemic in urban settlements in Lesotho, South Africa and Zimbabwe.”

### 2.1. Design, Population and Sampling

The study comprised a quantitative pre-test, post-test design. The study population included Rampepe Village in the Leribe settlement in Lesotho. All households in this area were eligible to participate in the Society of Women against AIDS in Africa (SWAALES) vegetable gardening program (referred to as the intervention partner). Following approval of the protocol by the Health Sciences Research Ethics Committee of the University of the Free State (ECUFS NO 94/2014) and the Ministry of Health in Lesotho, a rapid appraisal census was undertaken in all households that were included in the list of beneficiaries of the intervention partner. During the census, information about the presence of a vegetable garden at each household was collected. Based on this information, households that met the inclusion criteria were identified. From the households included in the census, 50 households that met the inclusion criteria were purposively selected. From the list of 50 households, 25 households were randomly included in the intervention group and 25 in the control group.

Inclusion criteria included:Households providing informed consent after being adequately informed;Households situated in the beneficiary community;Households with no physical evidence of a garden;

Households with a non-functioning garden—there may have been a garden, but the garden was not being maintained and had not produced any crops during the past season.

### 2.2. Data Collection Procedures

Prior to the baseline household interviews, fieldworkers from the National University of Lesotho received training on the data collection process. Measures of food security included the Living Poverty Index (LPI), Months of Adequate Household Food Provisioning (MAHFP), Household Dietary Diversity (HDD), and frequency of vegetables eaten.

A pilot study was undertaken prior to the commencement of the data collection phase of the study to determine the amount of time needed to complete the questionnaires as well as the clarity of the questions. Changes that were made to questionnaires following the pilot study included the following:Underlining of key words in order to assist the interviewer;Instructions for the interviewer at certain questions were inserted to ensure uniformity;Wording/phrasing of questions and answer options were changed in the questionnaire to make them more applicable to the target population.

The LPI scale was used [21] to assess the standard of living (an indirect indicator of both poverty and food insecurity) by determining the frequency that households went without basic necessities of life (namely food, water, medicine, electricity and fuel, and cash income). It was assessed for a period of 12 months prior to the interview using a set of six questions, each with six possible responses (the sixth response being ‘I don’t know’). The range of responses each received a score on a five-point scale. The responses were then combined to calculate the average LPI score for the household, 0 (no poverty) to 4 (complete poverty) [21].

The standardized MAFHP questionnaire developed by Billinsky and Swindale, another indirect measure of food security, was applied to determine MAHFP during the previous year [22]. For each household, MAHFP was calculated by subtracting the total number of months out of the previous 12 months that the household was unable to meet their food needs from 12 (e.g., 12–sum (A + B + C + D + E + F + G + H + I + J + K + L). Values for A through L were ‘0′ or ‘1′ [22]. The scores were categorized into three groups according to level of food security. A score of 12 meant that the household had year-round adequate food provisioning. Households that scored between 11 and 8 were deemed to have moderate food security, households that scored between 4 and 7 low level of food security, and households that scored between zero and 3 were considered to be food-insecure.

HDD Score (HDDS) was defined as the number of food groups consumed during the previous day. The level of diversity in the household diet was determined using the standardized questionnaire on HDD [23] after which the HDDS of a household was calculated. In this questionnaire, the number of different food groups consumed during the previous day was determined from a possible 12 food groups, which included cereals; roots and tubers; vegetables; fruits; meat, poultry, offal; eggs; fish, and seafood; pulses/legumes/nuts; milk and milk products; oil/fats; sugar/honey; and miscellaneous. Once the data had been obtained, the HDDS was calculated by tallying the total number of food groups from 12 consumed by the members of the household. The HDDS were interpreted in the following way: 0–3 = low dietary diversity; 4–5 = medium dietary diversity, and 6–12 = high dietary diversity [23].

Once the control and intervention households were identified, the first set of baseline interviews was undertaken. A two-person team of fieldworkers, from the National University of Lesotho, conducted structured interviews with the head of the household at the household.

### 2.3. Implementation of the Intervention

As mentioned, the households that were included in the household food garden intervention were beneficiaries of the SWAALES. SWAALES is a nonprofit, nongovernmental organization that aims to achieve a HIV/AIDS-free world and to empower African Women and Children to claim equal rights, access to healthcare, education, economic, and sociocultural opportunities. The 25 intervention households were trained on and assisted with the implementation of their household food gardens from July to September 2014. Vegetables that were planted for the summer harvest included pumpkin, carrots, spinach, green beans, tomatoes, onion, beetroot, and potatoes. Planting for the winter harvest took place between January and June 2015. The gardeners within those households received basic training on garden layout and bed design; natural soil fertility, pest and disease control; food preservation, processing, and storing; seed harvesting and saving; and preparation for winter crops, frost, and cold damage. Training of beneficiary households of the SWAALES program was done by Lima Rural Development Foundation representatives. Maintenance and monitoring continued throughout the remainder of the project (July 2014 to September 2015). Training material(s) were made available in Sesotho. Control households were informed of the incentive they would receive at the completion of the study (the same household food garden intervention as the intervention group).

Following the implementation of the household food garden intervention, a follow-up survey took place between July and September of 2015. The same team that had conducted the baseline survey conducted the follow-up interviews using the same questionnaire.

### 2.4. Validity and Reliability

Validity was guaranteed by researching evidence-based literature concerning indirect measures of household food security and including these in the questionnaire. To ensure reliability, the same questionnaires was used to obtain information from participants. Structured interviews were conducted by local fieldworkers who received training on research ethics, survey methodology, and fieldwork. The use of structured interviews contributed to reliability as they helped eliminate the possibility of research bias and subjectivity, since questions were asked exactly as worded in the questionnaire. Fieldworkers were assisted by footnote instructions in the questionnaire guiding them throughout the interview. The order in which the questions were asked was also carefully considered to avoid previous questions influencing the participant’s answers. The questionnaires were translated into the local language, with the original translations being back-translated by another person to improve the reliability of the translations.

### 2.5. Statistical Analysis

Data were captured using a double entry process in Census and Survey Processing System (CSPro) software (Washington DC, USA). An exact replica of the questionnaire was programmed into the CSpro programme and was programmed to disallow any irrelevant data entry (e.g., words instead of numbers, incorrect values), enabling the data capturers to maintain the integrity of the data. Descriptive statistics, namely frequencies and percentages for categorical data and medians and ranges for numerical data, were calculated before and after the intervention per group. The changes from before to after the intervention were calculated and described by means of 95% confidence intervals (CIs).

## 3. Results

### 3.1. Household Demographics, Responsibilities, and Structure

No significant differences occurred between the control and intervention groups at baseline in terms of household demographics (Table 1). Gender distribution in the control and intervention group was 72% male and 28% female and 64% male and 36% female, respectively. In both groups, about half of participants were married (Control: 48%; Intervention: 56%), followed by individuals who were separated (Control 32%; Intervention 44%), and slightly more individuals who were unmarried and widowed in the control group (Control 12%; Intervention 0%). In terms of education, about 40% of participants had completed primary school (Control 44%; Intervention 41.7%), while a higher percentage of participants in the intervention group had completed some high school (Control: 20.0%; Intervention: 41.7%).

Households in both groups (about 1 in 4) acknowledged a female as the head of the household (Control 24%; Intervention 24%), while a similar percentage acknowledged a male-headed household (Control 24%; Intervention 16%). At baseline, the main meal of the day was eaten at home by about 90% of all participants (Control 88%; Intervention 100%). In both groups, household responsibilities such as buying food were mainly the responsibility of the head of the household (Control 68%; Intervention 64%), who was also the most likely to decide who receives food and when (Control 60%; Intervention 52%).

### 3.2. Living Poverty Index

Table 2 indicates the results pertaining to the categories of LPI of the control and intervention groups.

The median LPI of 2.5 (range 1.7–4.5) in the control group and 2.83 (range 1.7–4.3) in the intervention group indicated a high level of poverty. The control and intervention groups were not different, except for the variable ‘enough clean water for the house.’ At baseline, 92% of the participants in the control group reported that they had never or just once or twice gone without water over the past 12 months, compared to 56% in the intervention group, a difference that was statistically significant (95% CI for the percentage difference (1.5%; 44.4%). At baseline, 44% of control households and 64% of intervention households reported not having enough food to eat ‘several and many times,’ with households that reported going without electricity ‘many times’ being similar in the two groups (Control: 100%; Intervention: 96%). Households that reported the absence of enough fuel to cook their food ‘several or many times’ were similar in the control (24%) and intervention (20%) groups, as were the percentage of households that went without a cash income ‘several or many times’ (60% in both the Control and Intervention groups).

### 3.3. Months of Adequate Household Food Provisioning

In Table 3, the percentage of participants that experienced adequate household food provisioning during the different months is depicted.

The descriptive data in Table 3 was used to determine the categories of scores for MAHFP (Table 4).

In terms of score categories for MAHFP, there were no significant differences in the percentage of respondents from scores in the different categories in the two groups at baseline and at follow-up. The households were grouped into food-secure categories according to their MAHFP score. Table 4 shows that at baseline, the control group did not have any households that scored in the very low and low category (0%), with the intervention group having only 8% of households in the low food security category. At follow-up, though not significant, a small improvement was noted in the intervention group where there were no longer any households in the low food security category.

In the control group, the median MAHFP score remained the same (11), while the median score in the intervention group improved by one point (from 10 to 11), a change that was not statistically significant (95% CI for the difference (−2; 0)).

### 3.4. Household Dietary Diversity

Table 5 shows the percentage of respondents that ate the indicated food groups during the previous day, while the HDD scores are depicted in Table 6.

The results from the DD data of the current study showed that, as expected, almost all participants consumed starchy cereals on a daily basis. The percentage of participants who consumed dairy and flesh meat every day was also high. After the intervention, the intake of most healthy food groups remained unchanged. Although slight improvements were seen in the intake of vitamin A-rich vegetables and other vegetables in the intervention group, the percentage of households that consumed these foods on a daily basis was still low. Furthermore, a large percentage of participants reported daily intake of unhealthy foods such as fats, oils, and sweets (at follow-up, about 70% of all participants reported consuming sweets on a daily basis). These results were used to categorize households as having low, medium, or high dietary diversity (Table 6).

At baseline, about one-third of the control and intervention households had a low level of dietary diversity (Control 36%; Intervention 28%), while half of the households in both the control and intervention group had a medium level of dietary diversity (48%). At follow-up, the percentage of households with a low HDD improved from 28% to 12% in the intervention group, while 40% of control households were categorized as having a low HDD, a difference that was statistically significant (95% CI for difference (2.5%; 50.7%)).

## 4. Discussion

The sociodemographic characteristics of the intervention and control groups were similar and reflected those reported in the LDHS. High levels of poverty were identified, and although the majority of participants were educated beyond primary school level, few had completed high school.

The most basic cause of food insecurity is inadequate access to food as a result of poverty [7,17,24]. According to the United Nations (UN), poverty is defined as “a human condition characterized by the sustained or chronic deprivation of the resources, capabilities, choices, security and power necessary for the enjoyment of an adequate standard of living and other civil, cultural, economic, political and social rights” [25]. Poverty is therefore closely linked to food insecurity. In the current study, the median LPI scores confirmed a high level of poverty.

Although almost no households were scored very low or low using the MAHFP tool, less than half of households were categorized as food-secure. The MAHFP focuses on the perception of whether there is enough food to eat in the household—it does not assess the variety or quality of the food that is available. In this context, DD is universally recognized as a key component of variety and therefore a useful tool when assessing food security [26,27]). The diets of people living in developing countries often lack diversity, since they often survive on staple plant-based diets [17,28]. Almost all households in the current study reported consuming starchy cereal-based foods during the previous day. According to Akhter et al., “diet quality is an important determinant of the food and nutrition security of a population and is influenced by food availability, access, utilisation and affordability at both country and household level” [7]. Darmon and Drewnowski found that poor households chose food to satisfy hunger, selecting cheaper energy-dense foods that were high in fat and sugar rather than more nutritious foods such as fruits and vegetables [24]. In this context, household food gardens have the potential to influence food security through increasing the availability of vegetables, contributing to a more diversified diet and a higher consumption of nutritious food [16,17,18,19,20].

Although the reported frequency of vegetables eaten improved significantly in the intervention group (but not in the control group) and slight improvements were seen in the intake of vitamin A-rich vegetables and other vegetables in the intervention group, the percentage of households that consumed these foods on a daily basis was still largely inadequate. The harsh environmental conditions in Lesotho may have played a role in the limited success of the intervention [4].

Although DD is often used as an indicator of food security, care needs to be taken when interpreting this information. A higher DDS does not guarantee the consumption of a nutrient-dense, quality diet. Unhealthy foods that are high in cheap animal proteins, fats, refined cereals, and sugar are generally more energy-dense and affordable than healthier foods [29]. The results from South African studies have shown that a high DDS may be related to an increased intake of unhealthy foods, such as fast foods [30,31]. The study of Rothman et al. confirmed that women from both rural and urban households in Lesotho underwent a nutrition transition and consumed unhealthy foods such as refined starches, fatty, and sugary foods [9]. The results of the current study are consistent with these findings, with a large percentage of participants reporting daily intake of unhealthy foods such as fats, oils, and sweets. The intake of these high-energy but nutrient-poor foods is closely linked to both food insecurity and overweight and obesity [12].

In addition to the potential of vegetable gardening interventions to address food security and to serve as a source of income generation, they are also associated with social and emotional benefits [16,18]. The creation and maintenance of gardens can contribute to building resilience and a sense of community in both adults and children. Darby et al. (2020) reported that low-income gardeners are motivated by pleasure from the practice of gardening, while also reinforcing social connections and cultural traditions [16].

The small sample size, as well as the fact that the measures that were applied to measure food security focused more on the experiences of participants and on the types of foods that were eaten, are limitations of the study. No information on quantities of foods (especially quantities of vegetables) that were eaten was collected, making it difficult to accurately determine the adequacy of the diet. The inclusion of more than one measure of food security in the current study provided a holistic view of the situation in Lesotho, which is a strength of the study. Applying a variety of tools made it possible to evaluate the contribution of a number of variables to food security, since they focused on different components.

## 5. Conclusions

The households included in the current study were characterized by high levels of poverty. Despite this, some measures of food security showed that participants were not as food-insecure as expected. Although significant improvements were noted in the frequency of vegetables consumed in the intervention group that were not noted in the control group, the percentage of households that ate vegetables was still far from the ideal of 400 g per day recommended by the WHO [32]. Faber et al. noted that, for vegetable gardens to have a sustainable impact, access to high-quality natural resources is required [29]. The unfavorable agroecological conditions in Lesotho may have contributed to the limited impact of the current intervention. In view of this, it may be helpful to equip potential gardeners with knowledge on gardening practices that consider environmental challenges (e.g., drought, frost, etc.). Sharing and communal gardens may also benefit this type of community as individuals pool resources and combine knowledge. Ultimately, interventions that target the basic and underlying causes of poor food security, such as poor socioeconomic circumstances, should be prioritized. Moreover, we recommend empirical research to develop an econometric model to further elucidate the impact of household food garden interventions in a larger sample.

## Figures and Tables

**Table 1 ijerph-17-08625-t001:** Household demographics, responsibilities and structure of control and intervention groups at baseline.

Baseline					
	Control (*n* = 25)		Intervention (*n* = 25)		95% CI for Difference at Baseline
	*n*	%	*n*	%	
Gender					
Male	18	72.0	16	64.0	
Female	7	28.0	9	36.0	−31.8%; 17.1%
Marital Status					
Unmarried	3	12.0	0	0	−3.5%; 30.0%
Married	12	48.0	14	56.0	
Living together/cohabiting	1	4.0	0	0	
Divorced	1	4.0	0	0	
Separated	8	32.0	11	44.0	
Widowed	3	12.0	0	0	
Highest level of education					
No formal schooling	0	0	0	0	
Some primary	4	16.0	0	0	
Primary completed	11	44.0	10	41.7	
Some high school	5	20.0	10	41.7	
High school completed	5	20.0	3	12.5	−20.2%; 9.6%
Tertiary education	0	0	1	4.2	
Where was the main meal eaten yesterday?					
Home (this household)	22	88.0	25	100.0	−30.0%; 3.5%
Shared with others	1	4.0	0	0	
Workplace	1	4.0	0	0	
Did not eat a meal	1	4.0	0	0	
Household structure					
Female centered	6	24.0	6	24.0	−23.1%; 23.1%
Male centered	6	24.0	4	16.0	
Nuclear	5	20.0	4	16.0	
Extended	3	12.0	10	40.0	
Live alone	5	20.0	1	4.0	
Who in the household does the following?					
buys food					
Household head	17	68.0	16	64.0	−21.1%; 28.5%
Other	8	32.0	9	36.0	
prepares food					
Household head	14	56.0	10	40.0	−11.0%; 40.0%
Other	11	44.0	15	60.0	
decides who gets food and when					
Household head	15	60.0	13	52.0	−18.3%; 32.9%
Other	10	40.0	12	48.0	

**Table 2 ijerph-17-08625-t002:** Living Poverty Index of control and intervention groups at baseline.

Baseline					
	Control (*n* = 25)		Intervention (*n* = 25)		95% CI for Difference at Baseline
	*n*	%	*n*	%	
Over the past 12 months, how often, if ever, have you or your family (household) gone without enough food to eat?					
Never	5	20.0	5	20.0	−5.2%; 45.4%
Once or twice	8	32.0	3	12.0	
Several times	6	24.0	9	36.0	−43.3%; 6.9%
Many times	5	20.0	7	28.0	
Always	1	4.0	1	4.0	
Don’t know	0	0	0	0.0	
Over the past 12 months, how often, if ever, have you or your family (household) gone without enough clean water for the home?					
Never	18	72.0	13	52.0	1.5%; 44.4% *
Once or twice	5	20.0	4	4.0	
Several times	0	0	2	8.0	−44.4%; −1.5% *
Many times	2	8.0	6	24.0	
Always	0	0	0	0	
Don’t know	0	0	0	0	
Over the past 12 months, how often, if ever, have you or your family (household) gone without medicine or medicinal treatment?					
	*n* = 24		*n* = 25		
Never	14	58.3	9	36.0	−22.8%; 27.5%
Once or twice	2	8.3	7	28.0	
Several times	4	16.7	4	16.0	27.5%; 22.8%
Many times	4	16.7	5	20.0	
Always	0	0	0	0	
Don’t know	0	0	0	0	
Over the past 12 months, how often, if ever, have you or your family (household) gone without electricity in your home?					
Never	0	0	0	0	−13.3%; 13.3%
Once or twice	0	0	0	0	
Several times	0	0	1	4.0	−13.3%; 13.3%
Many times	25	100.0	24	96.0	
Always	0	0	0	0	
Don’t know	0	0	0	0	
Over the past 12 months, how often, if ever, have you or your family (household) gone without enough fuel to cook your food?					
Never	13	52.0	15	60.0	−31.2%; 16.4%
Once or twice	4	16.0	4	16.0	
Several times	4	16.0	3	12.0	−16.4%; 31.2%
Many times	2	8.0	2	8.0	
Always	2	8.0	1	4.0	
Don’t know	0	0	0	0	
Over the past 12 months, how often, if ever, have you or your household gone without a cash income?					
Never	1	4.0	1	4.0	15.5%; 30.5%
Once or twice	6	24.0	4	16.0	
Several times	11	44.0	6	24.0	−30.5%; 15.5%
Many times	4	16.0	9	36.0	
Always	3	12.0	5	20.0	
Don’t know	0	0	0	0	

* statistically significant.

**Table 3 ijerph-17-08625-t003:** Months of Adequate Household Food Provisioning of control and intervention groups at baseline and follow-up.

	Baseline				Follow-up			
	Control (*n* = 25)		Intervention (*n* = 25)		Control (*n* = 20)		Intervention (*n* = 25)	
	*n*	%	*n*	%	*n*	%	*n*	%
In the past 12 months, were there months in which you did not have enough food to meet your family’s needs?								
Yes	17	68	20	80	11	55	14	56
No	8	32	5	20	9	45	11	44
If Yes, which were the months (in the past 12 months) in which you did not have enough food to meet your family’s needs?								
January								
No	19	76	19	76	16	80	20	80
Yes	6	24	6	24	4	20	5	20
February								
No	21	84	20	80	16	80	20	80
Yes	4	16	5	20	4	20	5	20
March								
No	21	84	17	68	17	85	24	96
Yes	4	16	8	32	3	15	1	4
April								
No	21	84	18	72	20	100	24	96
Yes	4	16	7	28	0	0	1	4
May								
No	18	72	16	64	19	95	23	92
Yes	7	28	9	36	1	5	2	8
June								
No	23	92	18	72	19	95	21	84
Yes	2	8	7	28	1	5	4	16
July								
No	23	92	16	64	20	100	23	92
Yes	2	8	9	36	0	0	2	8
August								
No	24	96	19	76	18	90	23	92
Yes	1	4	6	24	2	10	2	8
September								
No	19	76	20	80	18	90	24	96
Yes	6	24	5	20	2	10	1	4
October								
No	24	96	20	80	18	90	24	96
Yes	1	4	5	20	2	10	1	4
November								
No	24	96	24	96	19	95	23	92
Yes	1	4	1	4	1	5	2	8
December								
No	22	88	24	96	18	90	21	84
Yes	3	12	1	4	2	10	4	16

**Table 4 ijerph-17-08625-t004:** Months of Adequate Household Food Provisioning (MAHFP) score categories of control and intervention groups at baseline and follow-up.

	Baseline					Follow Up				
Category	Control (*n* = 25)		Intervention (*n* = 25)		95%CI for % Difference	Control (*n* = 20)		Intervention (*n* = 25)		95%CI for % Difference
	*n*	%	*n*	%		*n*	%	*n*	%	
**Very low (0)**	0	0	0	0	−13.3%; 13.3%	0	0	0	0	−13.3%; 16.1%
**Low (1–4)**	0	0	2	8.0	−25.0%; 6.5%	0	0	0	0	−13.3%; 16.1%
**Moderate (5–8)**	3	12.0	4	16.0	−24.2%; 16.4%	1	5.0	1	4.0	−15.1%; 19.9%
**High (9–11)**	14	56.0	14	56.0	−25.7%; 25.7%	10	50.0	13	52.0	−28.9%; 25.3%
**Food-Secure (12)**	8	32.0	5	20.0	−12.2%; 34.5%	9	45.0	11	44.0	−26.0%; 28.1%

**Table 5 ijerph-17-08625-t005:** Household Dietary Diversity of control and intervention groups at baseline and follow-up.

	Baseline				Follow-Up			
	Control (*n* = 25)		Intervention (*n* = 25)		Control (*n* = 20)		Intervention (*n* = 25)	
	*n*	%	*n*	%	*n*	%	*n*	%
**Food Groups**								
Cereals	24	96	25	100	20	100	25	100
Vitamin A-rich vegetables	0	0	0	0	0	0	5	20
White vegetables and roots	4	16	1	4	0	0	2	8
Dark green leafy vegetables	8	32	7	28	5	25	7	28
Other vegetables	2	8	4	16	1	5	5	20
Vitamin A-rich fruit	0	0	0	0	0	0	1	4
Other fruit	1	4	2	8	1	5	0	0
Organ meat	1	4	1	4	0	0	1	4
Flesh meats	23	92	22	88	17	85	23	92
Eggs	4	16	7	28	1	5	5	20
Fish	0	0	0	0	0	0	3	12
Legumes, nuts, and seeds	2	8	5	20	4	20	4	16
Milk and milk products	21	84	23	92	19	95	22	88
Oils and fats	16	64	13	64	10	50	21	84
Sweets	13	52	10	40	14	70	17	68
Spices, condiments, and beverages	9	36	7	28	9	45	4	16

**Table 6 ijerph-17-08625-t006:** Household Dietary Diversity Scores of control and intervention groups at baseline and follow-up.

	Baseline					Follow Up				
Category	Control (*n* = 25)		Intervention (*n* = 25)		95% CI for % Difference	Control (*n* = 20)		Intervention (*n* = 25)		95% CI for % Difference
	*n*	%	*n*	%		*n*	%	*n*	%	
**Low (0–3)**	9	36.0	7	28.0	−17.1%; 31.8%	8	40.0	3	12.0	2.5%; 50.7% *
**Medium (4–5)**	12	48.0	12	48.0	−25.8%; 25.8%	11	55.0	17	68.0	−38.5%; 14.4%
**High (6–12)**	4	16.0	6	24.0	−29.7%; 14.5%	1	5.0	5	2.0	−34.5%;6.7%

* Statistically significant difference.

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
