# Peer review of "The Impact of a Household Food Garden Intervention on Food Security in Lesotho"

_ijerph, 2020, doi:10.3390/ijerph17228625_

Round 1

Reviewer 1 Report

- The keywords should not overlap with the title

- The introduction provide sufficient background.

- The research procedure is well described, but its structure needs to be improved (i.e. the content could be divided into subsections or to insert a figure). Furthermore, it is necessary to explain better the difference between the control and intervention households.

- Tables should be self-explanatory.

- L. 132: “developed by Billinsky and Swindale was applied” - cite the appropriate author and year as usual.

- L. 323: “[…] the percentage of households that ate vegetables was still far from ideal” – what is an ideal quantity? This discussion should be included in the manuscript, even though a limitation of your study was to collect data on the frequency, but not on the weight of food consumed.

- Conclusions should be supported by the results. You should use your data in order to improve this section.

- Since there are several abbreviations, consider also giving a list of abbreviations at the end of the manuscript: MAHFP, CI, HDD, HDDS, MPI, DD, DDS…

- There are some typing mistakes

Author Response

Comment: The keywords should not overlap with the title.

Reply: Keywords replaced with: Living Poverty Index, Adequate Household Food Provisioning, Dietary Diversity

Comment: The introduction provide sufficient background.

Reply: Thank you

Comment: The research procedure is well described, but its structure needs to be improved (i.e. the content could be divided into subsections or to insert a figure).

Reply: I initially removed sub-headings in order to adhere to the journal guidelines (template), but I agree that subheadings improve the structure – I have added the following sub-headings:

  • Design, population and sampling;
  • Data collection procedures;
  • Implementation of the intervention;
  • Validity and reliability;
  • Statistical analysis.

Comment: Tables should be self-explanatory.

Reply: Table headings amended to read as follows:

Table 1: Household demographics, responsibilities and structure of control and intervention groups at baseline

Table 2. Living Poverty Index of control and intervention groups at baseline

Table 3. Months of Adequate Household Food Provisioning of control and intervention groups at baseline and follow-up

Table 4. MAHFP Score categories of control and intervention groups at baseline and follow-up

Table 5. Household Dietary Diversity of control and intervention groups at baseline and follow-up

Table 6. Household Dietary Diversity Scores of control and intervention groups at baseline and follow-up

Comment:- L. 132: “developed by Billinsky and Swindale was applied” - cite the appropriate author and year as usual.

Reply: I have added the citation at the end of the sentence.

Comment:- L. 323: “[…] the percentage of households that ate vegetables was still far from ideal” – what is an ideal quantity? This discussion should be included in the manuscript, even though a limitation of your study was to collect data on the frequency, but not on the weight of food consumed.

Reply: I have added the following to the sentence:

……the percentage of households that ate vegetables was still far from the ideal of 400g per day recommended by the WHO [32].

Comment:  Conclusions should be supported by the results. You should use your data in order to improve this section.

Reply: Amended to read as follows:

The households included in the current study were characterized by high levels of poverty. Despite this, some measures of food security showed that participants were not as food insecure as one would have expected. Although significant improvements were noted in the frequency of vegetables consumed in the intervention group that were not noted in the control group, the percentage of households that ate vegetables was still far from the ideal of 400g per day recommended by the WHO [32]. Faber et al. have noted that for vegetable gardens to have a sustainable impact, access to high quality natural resources is required [2929]. The unfavorable agro-ecological conditions in Lesotho may have contributed to the limited impact of the current intervention.

Comment:- Since there are several abbreviations, consider also giving a list of abbreviations at the end of the manuscript: MAHFP, CI, HDD, HDDS, MPI, DD, DDS…

Reply: Added as follows:

List of abbreviations:

CI: Confidence interval

DD: Dietary Diversity

HDD: Household Dietary Diversity

HDDS: Household Dietary Diversity Score

LDHS: Lesotho Demographic and Health Survey

LPI: Living Poverty Index

MAHFP: Months of Adequate Household Food Provisioning

SWAALES: Society of Women against AIDS in Africa

Comment: There are some typing mistakes

Reply: The whole manuscript has been edited again and changes are indicated in track changes.

Reviewer 2 Report

Well-written and clearly argued article. You have successfully made a case for the proposed program and provided sufficient data to substantiate its relevance and value. A definition of 'stunted' and 'wasted' in terms of children's development is missing from the article and the abbreviation DD when presented for the first time should be accompanied by the full terms. You could improve the narrative- the text is too dense and descriptive. It would also be beneficial to present similar approaches and their outcomes from relevant contexts. In your discussion you should  stress the program's impact on social sustainability, emotional benefits and resilience, and community building. Finally, you mention the importance of the program for children but it does not involve them into the creation and maintenance of the gardens. Overall, a very interesting project. 

Author Response

Well-written and clearly argued article. You have successfully made a case for the proposed program and provided sufficient data to substantiate its relevance and value.

Comment: A definition of 'stunted' and 'wasted' in terms of children's development is missing from the article

Reply: Added as follows:

The most recent Lesotho Demographic and Health Survey (LDHS) confirmed that a triple burden of malnutrition, characterized by undernutrition (stunting, defined as height-for-age below -2 standard deviations from the WHO mean; and wasting, defined as weight-for-height below -2 standard deviations from the WHO mean), micro-nutrient deficiency and overnutrition (overweight and obesity) was evident in 2013 [2]. 

Comment: the abbreviation DD when presented for the first time should be accompanied by the full terms.

Reply: Amended (marked in track changes in line 87)

Comment: You could improve the narrative- the text is too dense and descriptive.

Reply: Changes have been marked throughout in track changes – to address this comment I have removed paragraphs of text (I hope that it does not affect the description of the context in Lesotho negatively – please advise).

Comment: It would also be beneficial to present similar approaches and their outcomes from relevant contexts.

Reply: More detail on the findings of the following three studies have been added:

Darby, K. J.; Hinton, T.; Torre, J.  The motivations and needs of rural, low-income household food gardeners. Journal of Agriculture, Food Systems, and Community Development 2020, 9(2), 55–69. DOI: 10.5304/jafscd.2020.092.002

Rammohan, A.; Pritchard, B.; Dibley M. Home gardens as a predictor of enhanced dietary diversity and food security in rural Myanmar. BMC Public Health 2019, 19,1145. DOI: 10.1186/s12889-019-7440-7

Porter, C. M. Growing our own: Characterizing food-production strategies with five U.S. community-based food justice organizations. Journal of Agriculture, Food Systems, and Community Development 2018, 8(Suppl. 1), 187–205. DOI: 10.5304/jafscd.2018.08A.002

Comment: In your discussion you should stress the program's impact on social sustainability, emotional benefits and resilience, and community building.

Reply: Supporting statements to address this comment have been added in track changes (I have used the previous three studies in this regard). I have added the following paragraph:

“In addition to the potential of vegetable gardening interventions to address food security and to serve as a source of income generation, they are also associated with social and emotional benefits [16, 18]. The creation and maintenance of gardens can contribute to building resilience, and a sense of community in both adults and children. Darby et al (2020) have reported that low-income gardeners are motivated by pleasure from the practice of gardening, while also reinforcing social connections and cultural traditions [16].”

Comment: Finally, you mention the importance of the program for children but it does not involve them into the creation and maintenance of the gardens.

Reply: I have removed the statistics on child undernutrition in Lesotho (also to make the text less dense as recommended). In the discussion section I have made brief mention of the benefits for children in terms of the creation and maintenance of the gardens – please see previous paragraph.

Overall, a very interesting project.

Reviewer 3 Report

This is an interesting study. The issue of food security is very important for developing countries. This study discusses the impact of household food garden intervention. This research is fluent in writing and can attract readers widely. It is recommended to publish after minor revisions. The specific suggestions are as follows.

(1) The authors selected the 25-household experimental group and the 25-household control group. In order to show that the intervention measures are effective, it is necessary to provide the basic information difference analysis of the two groups. For example, add some T-test results to table1.

(2) The statistical results can help us understand the data. However, empirical research is also essential. It is recommended that the author add a econometric model to discuss the impact of household food garden intervention.

Author Response

This is an interesting study. The issue of food security is very important for developing countries. This study discusses the impact of household food garden intervention. This research is fluent in writing and can attract readers widely. It is recommended to publish after minor revisions. The specific suggestions are as follows.

Comment: (1) The authors selected the 25-household experimental group and the 25-household control group. In order to show that the intervention measures are effective, it is necessary to provide the basic information difference analysis of the two groups. For example, add some T-test results to table1.

Reply: Thank you for your comment. Although we have not indicated p-values, we have included CIs to show the statistical significance of differences between groups.  When a confidence interval excludes zero, it means that the difference is statistically significant.  Thus, the researcher is 95% confident that the difference that would be found in the population would be greater than zero, which is equivalent to rejecting the null hypothesis of no difference at P <.05 level by means of the applicable statistical such as a t-test or Mann-Whitney test.

References

Sim J, Reid N. Statistical inference by confidence intervals: Issues of interpretation and utilization. 1999. Physical Therapy 29(2):186-195

Fethney J. 2010. Statistical and clinical significance, and how to use confidence intervals to help interpret both.  Australian Critical Care. 23: 93-97

Comment: (2) The statistical results can help us understand the data. However, empirical research is also essential. It is recommended that the author add a econometric model to discuss the impact of household food garden intervention.

Reply: We did consider including our results in a model, but in our opinion, the sample size is too small to do a meaningful regression analysis. We have added this as a recommendation for future studies.

Round 2

Reviewer 1 Report

I consider that conclusion section should avoid references.
Reference from WHO is accepted, but reference from Faber et al. should be deleted in this section.

Conclusions should be supported by the results. You should use your data (numbers) in order to improve this section.